# Deep Reinforcement Learning in a Handful of Trials using Probabilistic Dynamics Models

Kurtland Chua      Roberto Calandra      Rowan McAllister      Sergey Levine

Berkeley Artificial Intelligence Research
University of California, Berkeley
{kchua, roberto.calandra, rmcallister, svlevine}@berkeley.edu

## Abstract

Model-based reinforcement learning (RL) algorithms can attain excellent sample efficiency, but often lag behind the best model-free algorithms in terms of asymptotic performance. This is especially true with high-capacity parametric function approximators, such as deep networks. In this paper, we study how to bridge this gap, by employing uncertainty-aware dynamics models. We propose a new algorithm called probabilistic ensembles with trajectory sampling (PETS) that combines uncertainty-aware deep network dynamics models with sampling-based uncertainty propagation. Our comparison to state-of-the-art model-based and model-free deep RL algorithms shows that our approach matches the asymptotic performance of model-free algorithms on several challenging benchmark tasks, while requiring significantly fewer samples (e.g., 8 and 125 times fewer samples than Soft Actor Critic and Proximal Policy Optimization respectively on the half-cheetah task).

## 1 Introduction

Reinforcement learning (RL) algorithms provide for an automated framework for decision making and control: by specifying a high-level objective function, an RL algorithm can, in principle, automatically learn a control policy that satisfies this objective. This has the potential to automate a range of applications, such as autonomous vehicles and interactive conversational agents. However, current model-free reinforcement learning algorithms are quite data-expensive to train, which often limits their application to simulated domains [Mnih et al., 2015, Lillicrap et al., 2016, Schulman et al., 2017], with a few exceptions [Kober and Peters, 2009, Levine et al., 2016]. A promising direction for reducing sample complexity is to explore model-based reinforcement learning (MBRL) methods, which proceed by first acquiring a predictive model of the world, and then using that model to make decisions [Atkeson and Santamaría, 1997, Kocijan et al., 2004, Deisenroth et al., 2014]. MBRL is appealing because the dynamics model is reward-independent and therefore can generalize to new tasks in the same environment, and it can easily benefit from all of the advances in deep supervised learning to utilize high-capacity models. However, the asymptotic performance of MBRL methods on common benchmark tasks generally lags behind model-free methods. That is, although MBRL methods tend to learn more quickly, they also tend to converge to less optimal solutions.

In this paper, we take a step toward narrowing the gap between model-based and model-free RL methods. Our approach is based on several observations that, though relatively simple, are critical for good performance. We first observe that model capacity is a critical ingredient in the success of MBRL methods: while efficient models such as Gaussian processes can learn extremely quickly, they struggle to represent very complex and discontinuous dynamical systems [Calandra et al., 2016]. By contrast, neural network (NN) models can scale to large datasets with high-dimensional inputs, and can represent such systems more effectively. However, NNs struggle with the opposite problem:

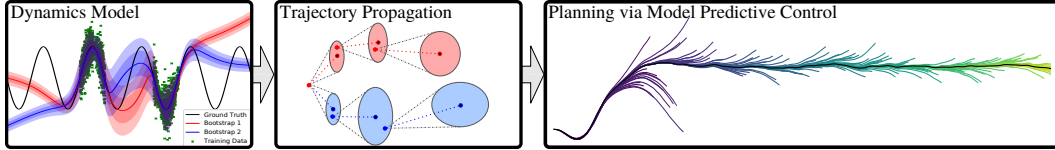

Figure 1: Our method (PE-TS): **Model**: Our probabilistic ensemble (PE) dynamics model is shown as an ensemble of two bootstraps (bootstrap disagreement far from data captures epistemic uncertainty: our subjective uncertainty due to a lack of data), each a probabilistic neural network that captures aleatoric uncertainty (inherent variance of the observed data). **Propagation**: Our trajectory sampling (TS) propagation technique uses our dynamics model to re-sample each particle (with associated bootstrap) according to its probabilistic prediction at each point in time, up until horizon $T$. **Planning**: At each time step, our MPC algorithm computes an optimal action sequence, applies the first action in the sequence, and repeats until the task-horizon.

to learn fast means to learn with few data and NNs tend to overfit on small datasets, making poor predictions far into the future. For this reason, MBRL with NNs has proven exceptionally challenging.

Our second observation is that this issue can, to a large extent, be mitigated by properly incorporating uncertainty into the dynamics model. While a number of prior works have explored uncertainty-aware deep neural network models [Neal, 1995, Lakshminarayanan et al., 2017], including in the context of RL [Gal et al., 2016, Depeweg et al., 2016], our work is, to our knowledge, the first to bring these components together in a deep MBRL framework that reaches the asymptotic performance of state-of-the-art model-free RL methods on benchmark control tasks.

Our main contribution is an MBRL algorithm called probabilistic ensembles with trajectory sampling (PETS)[1] summarized in Figure 1 with high-capacity NN models that incorporate uncertainty via an ensemble of bootstrapped models, where each model encodes *distributions* (as opposed to point predictions), rivaling the performance of model-free methods on standard benchmark control tasks at a fraction of the sample complexity. An advantage of PETS over prior probabilistic MBRL algorithms is an ability to isolate two distinct classes of uncertainty: aleatoric (inherent system stochasticity) and epistemic (subjective uncertainty, due to limited data). Isolating epistemic uncertainty is especially useful for directing exploration [Thrun, 1992], although we leave this for future work. Finally, we present a systematic analysis of how incorporating uncertainty into MBRL with NNs affects performance, during both model training and planning. We show, that PETS' particular treatment of uncertainty significantly reduces the amount of data required to learn a task, e.g., eight times fewer data on half-cheetah compared to the model-free Soft Actor Critic algorithm [Haarnoja et al., 2018].

## 2   Related work

Model choice in MBRL is delicate: we desire effective learning in both low-data regimes (at the beginning) and high-data regimes (in the later stages of the learning process). For this reason, Bayesian nonparametric models, such as Gaussian processes (GPs), are often the model of choice in MBRL, especially in low-dimensional problems where data efficiency is critical [Kocijan et al., 2004, Ko et al., 2007, Nguyen-Tuong et al., 2008, Grancharova et al., 2008, Deisenroth et al., 2014, Kamthe and Deisenroth, 2018]. However, such models introduce additional assumptions on the system, such as the smoothness assumption inherent in GPs with squared-exponential kernels [Rasmussen and Kuss, 2003]. Parametric function approximators have also been used extensively in MBRL [Hernandaz and Arkun, 1990, Miller et al., 1990, Lin, 1992, Draeger et al., 1995], but were largely supplanted by Bayesian models in recent years. Methods based on local models, such as guided policy search algorithms [Levine et al., 2016, Finn et al., 2016, Chebotar et al., 2017], can efficiently train NN policies, but use time-varying linear models, which only locally model the system dynamics. Recent improvements in parametric function approximators, such as NNs, suggest that such methods are worth revisiting [Baranes and Oudeyer, 2013, Fu et al., 2016, Punjani and Abbeel, 2015, Lenz et al., 2015, Agrawal et al., 2016, Gal et al., 2016, Depeweg et al., 2016, Williams et al., 2017, Nagabandi et al., 2017]. Unlike Gaussian processes, NNs have constant-time inference and tractable training in the large data regime, and have the potential to represent more complex functions, including non-

smooth dynamics that are often present in robotics [Fu et al., 2016, Mordatch et al., 2016, Nagabandi et al., 2017]. However, most works that use NNs focus on deterministic models, consequently suffering from overfitting in the early stages of learning. For this reason, our approach is able to achieve even higher data-efficiency than prior deterministic MBRL methods such as Nagabandi et al. [2017].

Constructing good Bayesian NN models remains an open problem [MacKay, 1992, Neal, 1995, Osband, 2016, Guo et al., 2017], although recent promising work exists on incorporating dropout [Gal et al., 2017], ensembles [Osband et al., 2016, Lakshminarayanan et al., 2017], and $\alpha$-divergence [Hernández-Lobato et al., 2016]. Such probabilistic NNs have previously been used for control, including using dropout [Gal et al., 2016, Higuera et al., 2018] and $\alpha$-divergence [Depeweg et al., 2016]. In contrast to these prior methods, our experiments focus on more complex tasks with challenging dynamics – including contact discontinuities – and we compare directly to prior model-based and model-free methods on standard benchmark problems, where our method exhibits asymptotic performance that is comparable to model-free approaches.

# 3   Model-based reinforcement learning

We now detail the MBRL framework and the notation used. Adhering to the Markov decision process formulation [Bellman, 1957], we denote the state $s \in \mathbb{R}^{d_s}$ and the actions $a \in \mathbb{R}^{d_a}$ of the system, the reward function $r(s, a)$, and we consider the dynamic systems governed by the transition function $f_{\theta} : \mathbb{R}^{d_s + d_a} \mapsto \mathbb{R}^{d_s}$ such that given the current state $s_t$ and current input $a_t$, the next state $s_{t+1}$ is given by $s_{t+1} = f(s_t, a_t)$. For probabilistic dynamics, we represent the conditional distribution of the next state given the current state and action as some parameterized distribution family: $f_{\theta}(s_{t+1}|s_t, a_t) = \Pr(s_{t+1}|s_t, a_t; \theta)$, overloading notation. Learning forward dynamics is thus the task of fitting an approximation $\widetilde{f}$ of the true transition function $f$, given the measurements $\mathcal{D} = \{(s_n, a_n), s_{n+1}\}_{n=1}^{N}$ from the real system.

Once a dynamics model $\widetilde{f}$ is learned, we use $\widetilde{f}$ to predict the distribution over state-trajectories resulting from applying a sequence of actions. By computing the expected reward over state-trajectories, we can evaluate multiple candidate action sequences, and select the optimal action sequence to use. In Section 4 we discuss multiple methods for modeling the dynamics, and in Section 5 we detail how to compute the distribution over state-trajectories given a candidate action sequence.

# 4   Uncertainty-aware neural network dynamics models

This section describes several ways to model the task's true (but unknown) dynamic function, including our method: an ensemble of bootstrapped probabilistic neural networks. Whilst uncertainty-aware dynamics models have been explored in a number of prior works [Deisenroth et al., 2014, Gal et al., 2016, Depeweg et al., 2016], the particular details of the implementation and design decisions in regard incorporation of uncertainty

Table 1: Model uncertainties captured.

| Model | Aleatoric uncertainty | Epistemic uncertainty |
|---|---|---|
| *Baseline Models* | | |
| Deterministic NN (D) | No | No |
| Probabilistic NN (P) | Yes | No |
| Deterministic ensemble NN (DE) | No | Yes |
| Gaussian process baseline (GP) | Homoscedastic | Yes |
| *Our Model* | | |
| Probabilistic ensemble NN (PE) | **Yes** | **Yes** |

have not been rigorously analyzed empirically. As a result, prior work has generally found that expressive parametric models, such as deep neural networks, generally do not produce model-based RL algorithms that are competitive with their model-free counterparts in terms of asymptotic performance [Nagabandi et al., 2017], and often even found that simpler time-varying linear models can outperform expressive neural network models [Levine et al., 2016, Gu et al., 2016].

Any MBRL algorithm must select a class of model to predict the dynamics. This choice is often crucial for an MBRL algorithm, as even small bias can significantly influence the quality of the corresponding controller [Atkeson and Santamaría, 1997, Abbeel et al., 2006]. A major challenge is building a model that performs well in low and high data regimes: in the early stages of training, data is scarce, and highly expressive function approximators are liable to overfit; In the later stages of training, data is plentiful, but for systems with complex dynamics, simple function approximators might underfit. While Bayesian models such as GPs perform well in low-data regimes, they do not scale favorably

with dimensionality and often use kernels ill-suited for discontinuous dynamics [Calandra et al., 2016], which is typical of robots interacting through contacts.

In this paper, we study how expressive NNs can be incorporated into MBRL. To account for uncertainty, we study NNs that model two types of uncertainty. The first type, aleatoric uncertainty, arises from *inherent stochasticities* of a system, e.g. observation noise and process noise. Aleatoric uncertainty can be captured by outputting the parameters of a parameterized distribution, while still training the network discriminatively. The second type – epistemic uncertainty – corresponds to *subjective uncertainty* about the dynamics function, due to a lack of sufficient data to uniquely determine the underlying system exactly. In the limit of infinite data, epistemic uncertainty should vanish, but for datasets of finite size, subjective uncertainty remains when predicting transitions. It is precisely the subjective epistemic uncertainty which Bayesian modeling excels at, which helps mitigate overfitting. Below, we describe how we use combinations of 'probabilistic networks' to capture aleatoric uncertainty and 'ensembles' to capture epistemic uncertainty. Each combination is summarized in Table 1.

**Probabilistic neural networks (P)**    We define a *probabilistic* NN as a network whose output neurons simply parameterize a probability distribution function, capturing aleatoric uncertainty, and should not be confused with Bayesian inference. We use the negative log prediction probability as our loss function $\text{loss}_\text{P}(\boldsymbol{\theta}) = -\sum_{n=1}^{N} \log \widetilde{f}_{\boldsymbol{\theta}}(\boldsymbol{s}_{n+1}|\boldsymbol{s}_n, \boldsymbol{a}_n)$. For example, we might define our predictive model to output a Gaussian distribution with diagonal covariances parameterized by $\boldsymbol{\theta}$ and conditioned on $\boldsymbol{s}_n$ and $\boldsymbol{a}_n$, i.e.: $\widetilde{f} = \Pr(\boldsymbol{s}_{t+1}|\boldsymbol{s}_t, \boldsymbol{a}_t) = \mathcal{N}(\boldsymbol{\mu}_\theta(\boldsymbol{s}_t, \boldsymbol{a}_t), \boldsymbol{\Sigma}_\theta(\boldsymbol{s}_t, \boldsymbol{a}_t))$. Then the loss becomes

$$\text{loss}_\text{Gauss}(\boldsymbol{\theta}) = \sum_{n=1}^{N} [\boldsymbol{\mu}_{\boldsymbol{\theta}}(\boldsymbol{s}_n, \boldsymbol{a}_n) - \boldsymbol{s}_{n+1}]^\top \boldsymbol{\Sigma}_{\boldsymbol{\theta}}^{-1}(\boldsymbol{s}_n, \boldsymbol{a}_n)[\boldsymbol{\mu}_{\boldsymbol{\theta}}(\boldsymbol{s}_n, \boldsymbol{a}_n) - \boldsymbol{s}_{n+1}] + \log \det \boldsymbol{\Sigma}_{\boldsymbol{\theta}}(\boldsymbol{s}_n, \boldsymbol{a}_n). \quad (1)$$

Such network outputs, which in our particular case parameterizes a Gaussian distribution, models aleatoric uncertainty, otherwise known as heteroscedastic noise (meaning the output distribution is a function of the input). However, it does not model epistemic uncertainty, which cannot be captured with purely discriminative training. Choosing a Gaussian distribution is a common choice for continuous-valued states, and reasonable if we assume that any stochasticity in the system is unimodal. However, in general, any tractable distribution class can be used. To provide for an expressive dynamics model, we can represent the parameters of this distribution (e.g., the mean and covariance of a Gaussian) as nonlinear, parametric functions of the current state and action, which can be arbitrarily complex but deterministic. This makes it feasible to incorporate NNs into a probabilistic dynamics model even for high-dimensional and continuous states and actions. Finally, an under-appreciated detail of probabilistic networks is that their variance has *arbitrary* values for out-of-distribution inputs, which can disrupt planning. We discuss how to mitigate this issue in Appendix **??**.

**Deterministic neural networks (D)**    For comparison, we define a deterministic NN as a special-case probabilistic network that outputs delta distributions centered around point predictions denoted as $\widetilde{f}_{\boldsymbol{\theta}}(\boldsymbol{s}_t, \boldsymbol{a}_t)$: $\widetilde{f}_{\boldsymbol{\theta}}(\boldsymbol{s}_{t+1}|\boldsymbol{s}_t, \boldsymbol{a}_t) = \Pr(\boldsymbol{s}_{t+1}|\boldsymbol{s}_t, \boldsymbol{a}_t) = \delta(\boldsymbol{s}_{t+1} - \widetilde{f}_{\boldsymbol{\theta}}(\boldsymbol{s}_t, \boldsymbol{a}_t))$, trained using the MSE loss: $\text{loss}_D(\boldsymbol{\theta}) = \sum_{n=1}^{N} \|\boldsymbol{s}_{n+1} - \widetilde{f}_{\boldsymbol{\theta}}(\boldsymbol{s}_n, \boldsymbol{a}_n)\|$. Although MSE can be interpreted as $\text{loss}_\text{P}(\boldsymbol{\theta})$ with a Gaussian model of fixed unit variance, in practice this variance cannot be used for uncertainty-aware propagation, since it does not correspond to any notion of uncertainty (e.g., a deterministic model with infinite data would be adding variance to particles for no good reason).

**Ensembles (DE and PE)**    A principled means to capture epistemic uncertainty is with Bayesian inference. Whilst accurate Bayesian NN inference is possible with sufficient compute [Neal, 1995], approximate inference methods [Blundell et al., 2015, Gal et al., 2017, Hernández-Lobato and Adams, 2015] have enjoyed recent popularity given their simpler implementation and faster training times. Ensembles of bootstrapped models are even simpler still: given a base model, no additional (hyper-)parameters need be tuned, whilst still providing reasonable uncertainty estimates [Efron and Tibshirani, 1994, Osband, 2016, Kurutach et al., 2018]. We consider ensembles of $B$-many bootstrap models, using $\boldsymbol{\theta}_b$ to refer to the parameters of our $b^\text{th}$ model $\widetilde{f}_{\boldsymbol{\theta}_b}$. Ensembles can be composed of deterministic models (DE) or probabilistic models (PE) – as done by Lakshminarayanan et al. [2017] – both of which define predictive probability distributions: $\widetilde{f}_{\boldsymbol{\theta}} = \frac{1}{B}\sum_{b=1}^{B} \widetilde{f}_{\boldsymbol{\theta}_b}$. A visual example is provided in Appendix **??**. Each of our bootstrap models have their unique dataset $\mathbb{D}_b$, generated by

sampling (with replacement) $N$ times the dynamics dataset recorded so far $\mathbb{D}$, where $N$ is the size of $\mathbb{D}$. We found $B = 5$ sufficient for all our experiments. To validate the number of layers and neurons of our models, we can visualize one-step predictions (e.g. Appendix **??**).

# 5    Planning and control with learned dynamics

This section describes different ways uncertainty can be incorporated into planning using probabilistic dynamics models. Once a model $\widetilde{f}_{\boldsymbol{\theta}}$ is learned, we can use it for control by predicting the future outcomes of candidate policies or actions and then selecting the particular candidate that is predicted to result in the highest reward. MBRL planning in discrete time over long time horizons is generally performed by using the dynamics model to recursively predict how an estimated Markov state will evolve from one time step to the next, e.g.: $\boldsymbol{s}_{t+2} \sim \Pr(\boldsymbol{s}_{t+2}|\boldsymbol{s}_{t+1}, \boldsymbol{a}_{t+1})$ where $\boldsymbol{s}_{t+1} \sim \Pr(\boldsymbol{s}_{t+1}|\boldsymbol{s}_t, \boldsymbol{a}_t)$. When planning, we might consider each action $\boldsymbol{a}_t$ to be a function of state, forming a policy $\pi : \boldsymbol{s}_t \rightarrow \boldsymbol{a}_t$, a function to optimize. Alternatively, we can plan and optimize for a sequence of actions, a process called model predictive control (MPC) [Camacho and Alba, 2013]. We use MPC in our own experiments for several reasons, including implementation simplicity, lower computational burden (no gradients), and no requirement to specify the task-horizon in advance, whilst achieving the same data-efficiency as Gal et al. [2016] who used a Bayesian NN with a policy to learn the cart-pole task in 2000 time steps. Our full algorithm is summarized in Section 6.

Given the state of the system $\boldsymbol{s}_t$ at time $t$, the prediction horizon $T$ of the MPC controller, and an action sequence $\boldsymbol{a}_{t:t+T} \doteq \{\boldsymbol{a}_t, \dots, \boldsymbol{a}_{t+T}\}$; the *probabilistic* dynamics model $\widetilde{f}$ induces a distribution over the resulting trajectories $\boldsymbol{s}_{t:t+T}$. At each time step $t$, the MPC controller applies the first action $\boldsymbol{a}_t$ of the sequence of optimized actions $\arg\max_{\boldsymbol{a}_{t:t+T}} \sum_{\tau=t}^{t+T} \mathbb{E}_{\widetilde{f}}[r(\boldsymbol{s}_\tau, \boldsymbol{a}_\tau)]$. A common technique to compute the optimal action sequence is a random sampling shooting method, due to its parallelizability and ease of implementation. Nagabandi et al. [2017] use deterministic NN models and MPC with random shooting to achieve data efficient control in higher dimensional tasks than what is feasible for GPs to model. Our work improves upon Nagabandi et al. [2017]'s data efficiency in two ways: First, we capture uncertainty in modeling and planning, to prevent overfitting in the low-data regime. Second, we use CEM [Botev et al., 2013] instead of random-shooting, which samples actions from a distribution closer to previous action samples that yielded high reward.

Computing the expected trajectory reward using recursive state prediction in closed-form is generally intractable. Multiple approaches to approximate uncertainty propagation can be found in the literature [Girard et al., 2002, Quiñonero-Candela et al., 2003]. These approaches can be categorized by how they represent the state distribution: deterministic, particle, and parametric methods. Deterministic methods use the mean prediction and ignore the uncertainty, particle methods propagate a set of Monte Carlo samples, and parametric methods include Gaussian or Gaussian mixture models, etc. Although parametric distributions have been successfully used in MBRL [Deisenroth et al., 2014], experimental results [Kupcsik et al., 2013] suggest that particle approaches can be competitive both computationally and in terms of accuracy, without making strong assumptions about the distribution used. Hence, we use particle-based propagation, specifically suited to our PE dynamics model which distinguishes two types of uncertainty, detailed in Section 5.1. Unfortunately, little prior work has empirically compared the design decisions involved in choosing the particular propagation method. Thus, we compare against several baselines in Section 5.2. Visual examples are provided in Appendix **??**.

## 5.1    Our state propagation method: trajectory sampling (TS)

Our method to predict plausible state trajectories begins by creating $P$ particles from the current state, $\boldsymbol{s}_{t=0}^p = \boldsymbol{s}_0 \, \forall p$. Each particle is then propagated by: $\boldsymbol{s}_{t+1}^p \sim \widetilde{f}_{\boldsymbol{\theta}_{b(p,t)}}(\boldsymbol{s}_t^p, \boldsymbol{a}_t)$, according to a particular bootstrap $b(p,t) \, in\{1, \dots, B\}$, where $B$ is the number of bootstrap models in the ensemble. A particle's bootstrap index can potentially change as a function of time $t$. We consider two TS variants:

- **TS1** refers to particles uniformly re-sampling a bootstrap per time step. If we were to consider an ensemble as a Bayesian model, the particles would be effectively continually re-sampling from the approximate *marginal posterior* of plausible dynamics. We consider TS1's bootstrap re-sampling to place a soft restriction on trajectory multimodality: particles separation cannot be attributed to the *compounding* effects of differing bootstraps using TS1.

- **TS∞** refers to particle bootstraps never changing during a trial. An ensemble is a collection of plausible models, which together represent the *subjective* uncertainty in function space of the true dynamics function $f$, which we assume is time invariant. TS∞ captures such time invariance since each particle's bootstrap index is made consistent over time. An advantage of using TS∞ is that aleatoric and epistemic uncertainties are separable [Depeweg et al., 2018]. Specifically, aleatoric state variance is the average variance of particles of same bootstrap, whilst epistemic state variance is the variance of the average of particles of same bootstrap indexes. Epistemic is the 'learnable' type of uncertainty, useful for directed exploration [Thrun, 1992]. Without a way to distinguish epistemic uncertainty from aleatoric, an exploration algorithm (e.g. Bayesian optimization) might mistakingly choose actions with high predicted reward-variance 'hoping to learn something' when in fact such variance is caused by persistent and irreducible system stochasticity offering zero exploration value.

Both TS variants can capture multi-modal distributions and can be used with any probabilistic model. We found $P = 20$ and $B = 5$ sufficient in all our experiments.

### 5.2 Baseline state propagation methods for comparison

To validate our state propagation method, in the experiments of Section 7.2 we compare against four alternative state propagation methods, which we now discuss.

**Expectation (E)** To judge the importance of our TS method using multiple particles to represent a distribution we compare against the aforementioned deterministic propagation technique. The simplest way to plan is iteratively propagating the expected prediction at each time step (ignoring uncertainty) $\boldsymbol{s}_{t+1} = \mathbb{E}[\widetilde{f}_{\boldsymbol{\theta}}(\boldsymbol{s}_t, \boldsymbol{a}_t)]$. An advantage of this approach over TS is reduced computation and simple implementation: only a single particle is propagated. The main disadvantage of choosing E over TS is that small model biases can compound quickly over time, with no way to tell the quality of the state estimate.

**Moment matching (MM)** Whilst TS's particles can represent multimodal distributions, forcing a unimodal distribution via moment matching (MM) can (in some cases) benefit MBRL data efficiency [Gal et al., 2016]. Although unclear why, Gal et al. [2016] (who use Gaussian MM) hypothesize this effect may be caused by smoothing of the loss surface and implicitly penalizing multi-modal distributions (which often only occur with uncontrolled systems). To test this hypothesis we use Gaussian MM as a baseline and assume independence between bootstraps and particles for simplicity $\boldsymbol{s}_{t+1}^p \overset{iid}{\sim} \mathcal{N}\left(\mathbb{E}_{p,b}\left[\boldsymbol{s}_{t+1}^{p,b}\right], \mathbb{V}_{p,b}\left[\boldsymbol{s}_{t+1}^{p,b}\right]\right)$, where $\boldsymbol{s}_{t+1}^{p,b} \sim \widetilde{f}_{\boldsymbol{\theta}_b}(\boldsymbol{s}_t^p, \boldsymbol{a}_t)$. Future work might consider other distributions too, such as the Laplace distribution.

**Distribution sampling (DS)** The previous MM approach made a strong unimodal assumption about state distributions: the state distribution at each time step was re-cast to Gaussian. A softer restriction on multimodality – between MM and TS – is to moment match w.r.t. the bootstraps only (noting the particles are otherwise independent if $B = 1$). This means that we effectively smooth the loss function w.r.t. epistemic uncertainty only (the uncertainty relevant to learning), whilst the aleatoric uncertainty remains free to be multimodal. We call this method distribution sampling (DS): $\boldsymbol{s}_{t+1}^p \sim \mathcal{N}\left(\mathbb{E}_b\left[\boldsymbol{s}_{t+1}^{p,b}\right], \mathbb{V}_b\left[\boldsymbol{s}_{t+1}^{p,b}\right]\right)$, with $\boldsymbol{s}_{t+1}^{p,b} \sim \widetilde{f}_{\boldsymbol{\theta}_b}(\boldsymbol{s}_t^p, \boldsymbol{a}_t)$.

## 6 Algorithm summary

Here we summarize our MBRL method *PETS* in Algorithm 1. We use the PE model to capture heteroskedastic aleatoric uncertainty and heteroskedastic epistemic uncertainty, which the TS planning method was able to best use. To guide the random shooting method of our MPC algorithm, we found that the CEM method learned faster (as discussed in Appendix **??**).

---

**Algorithm 1** Our model-based MPC algorithm '*PETS*':

1: Initialize data $\mathbb{D}$ with a random controller for one trial.
2: **for** Trial $k = 1$ to $K$ **do**
3:     Train a *PE* dynamics model $\widetilde{f}$ given $\mathbb{D}$.
4:     **for** Time $t = 0$ to TaskHorizon **do**
5:         **for** Actions sampled $\boldsymbol{a}_{t:t+T} \sim \text{CEM}(\cdot)$, 1 to NSamples **do**
6:             Propagate state particles $\boldsymbol{s}_\tau^p$ using *TS* and $\widetilde{f}|\{\mathbb{D}, \boldsymbol{a}_{t:t+T}\}$.
7:             Evaluate actions as $\sum_{\tau=t}^{t+T} \frac{1}{P} \sum_{p=1}^{P} r(\boldsymbol{s}_\tau^p, \boldsymbol{a}_\tau)$
8:             Update CEM$(\cdot)$ distribution.
9:         Execute first action $\boldsymbol{a}_t^*$ (only) from optimal actions $\boldsymbol{a}_{t:t+T}^*$.
10:         Record outcome: $\mathbb{D} \leftarrow \mathbb{D} \cup \{\boldsymbol{s}_t, \boldsymbol{a}_t^*, \boldsymbol{s}_{t+1}\}$.

---

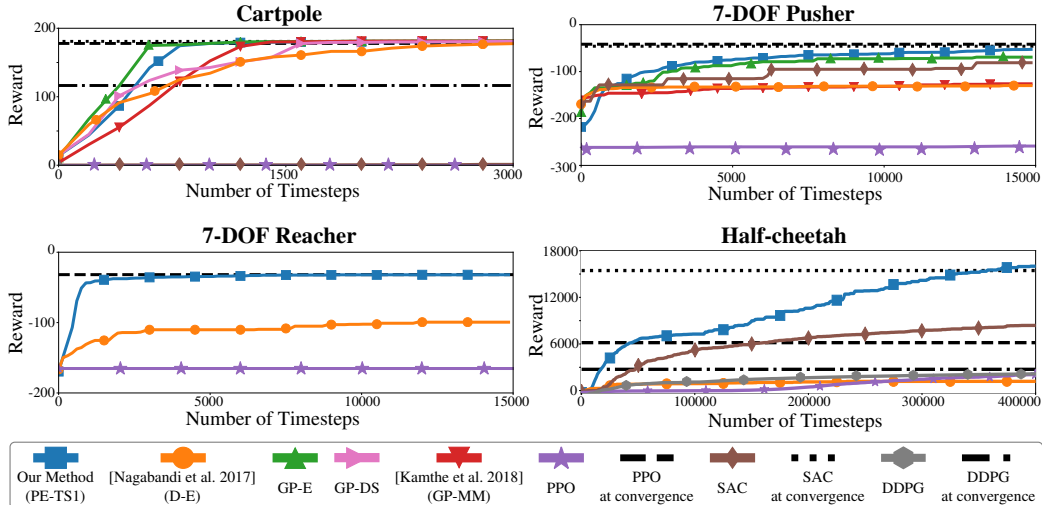

Figure 3: Learning curves for different tasks and algorithm. For all tasks, our algorithm learns in under 100K time steps or 100 trials. With the exception of Cartpole, which is sufficiently low-dimensional to efficiently learn a GP model, our proposed algorithm significantly outperform all other baselines. For each experiment, one time step equals 0.01 seconds, except Cartpole with 0.02 seconds. For visual clarity, we plot the average over 10 experiments of the maximum rewards seen so far.

# 7 Experimental results

We now evaluate the performance of our proposed MBRL algorithm called PETS using a deep neural network probabilistic dynamics model. First, we compare our approach on standard benchmark tasks against state-of-the-art model-free and model-based approaches in Section 7.1. Then, in Section 7.2, we provide a detailed evaluation of the individual design decisions in the model and uncertainty propagation method and analyze their effect on performance. Additional considerations of horizon length, action sampling distribution, and stochastic systems are discussed in Appendix **??**. The experiment setup is shown in Figure 2, and NN architecture details are discussed in the sup-

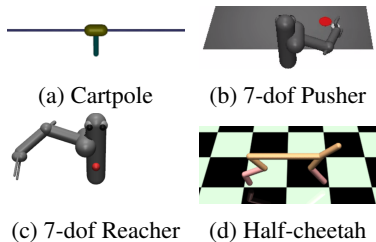

(a) Cartpole      (b) 7-dof Pusher

(c) 7-dof Reacher      (d) Half-cheetah

Figure 2: Tasks evaluated.

plementary materials, in Appendix **??**. Videos of the experiments, and code for reproducing the experiments can be found at `https://sites.google.com/view/drl-in-a-handful-of-trials`.

## 7.1 Comparisons to prior reinforcement learning algorithms

We compare our Algorithm 1 against the following reinforcement learning algorithms for continuous state-action control:

- **Proximal policy optimization (PPO)**: [Schulman et al., 2017] is a model-free, deep policy-gradient RL algorithm (we used the implementation from Dhariwal et al. [2017].)

- **Deep deterministic policy gradient (DDPG)**: [Lillicrap et al., 2016] is an off-policy model-free deep actor-critic algorithm (we used the implementation from Dhariwal et al. [2017].)

- **Soft actor critic (SAC)**: [Haarnoja et al., 2018] is a model-free deep actor-critic algorithm, which reports better data-efficiency than DDPG on MuJoCo benchmarks (we obtained authors' data).

- **Model-based model-free hybrid (MBMF)**: [Nagabandi et al., 2017] is a recent deterministic deep model-based RL algorithm, which we reimplement.

- **Gaussian process dynamics model (GP)**: we compare against three MBRL algorithms based on GPs. GP-E learns a GP model, but only propagate the expectation. GP-DS uses the propagation method DS. GP-MM is the algorithm proposed by Kamthe and Deisenroth [2018] except that we do *not* update the dynamics model after each transition, but only at the end of each trial.

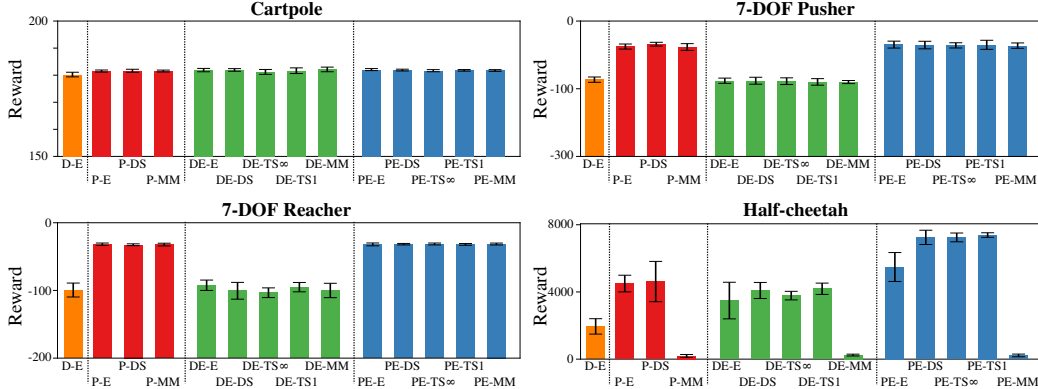

Figure 4: Final performance for different tasks, models, and uncertainty propagation techniques. The model choice seems to be more important than the technique used to propagate the state/action space. Among the models the ranking in terms of performance is: $PE > P > DE > D$. A linear model comparison can also be seen in Appendix **??**.

The results of the comparison are presented in Figure 3. Our method reaches performance that is similar to the asymptotic performance of the state-of-the-art model-free baseline PPO. However, PPO requires several orders of magnitude more samples to reach this point. We reach PPO's asymptotic performance in fewer than 100 trials on all four tasks, faster than any prior model-free algorithm, and the asymptotic performance substantially exceeds that of the prior MBRL algorithm by Nagabandi et al. [2017], which corresponds to the deterministic variant of our approach (D-E). This result highlights the value of uncertainty estimation. Moreover, the experiments show that NNs dynamics can achieve similar performance to GPs on low-dimensional tasks (i.e., cartpole), while also scaling to higher dimensional tasks such as half-cheetah. Whilst the probabilistic baseline GP-MM slightly outperformed our method in cartpole, GP-MM scales cubically in time and quadratically in state dimensionality, so was infeasible to run on the remaining higher dimensional tasks. It is worth noting that model-based deep RL algorithms have typically been considered to be efficient but incapable of achieving similar asymptotic performance as their model-free counterparts. Our results demonstrate that a purely model-based deep RL algorithm that only learns a dynamics model, omitting even a parameterized policy, can achieve comparable performance when properly incorporating uncertainty estimation during modeling and planning. In the next section, we study which specific design decisions and components of our approach are important for achieving this level of performance.

## 7.2 Analyzing dynamics modeling and uncertainty propagation

In this section, we compare different choices for the dynamics model in Section 4 and uncertainty propagation technique in Section 5. The results in Figure 4 first show that w.r.t. model choice, the model should consider both uncertainty types: the probabilistic ensembles (PE-XX) perform best in all tasks, except cartpole ('X' symbolizes any character). Close seconds are the single-probability-type models: probabilistic network (P-XX) and ensembles of deterministic networks (E-XX). Worst is the deterministic network (D-E).

These observations shed some light on the role of uncertainty in MBRL, particularly as it relates to discriminatively trained, expressive parametric models such as NNs. Our results suggest that, the quality of the model and the use of uncertainty at learning time significantly affect the performance of the MBRL algorithms tested, while the use of more advanced uncertainty propagation techniques seem to offers only minor improvements. We reconfirm that moment matching (MM) is competitive in low-dimensional tasks (consistent with [Gal et al., 2016]), however is not a reliable MBRL choice in higher dimensions, e.g. the half-cheetah.

The analysis provided in this section summarizes the experiments we conducted to design our algorithm. It is worth noting that the individual components of our method – ensembles, probabilistic networks, and various approximate uncertainty propagation techniques – have existed in various forms in supervised learning and RL. However, as our experiments here and in the previous section show, the particular choice of these components in our algorithm achieves substantially improved

results over previous state-of-the-art model-based and model-free methods, experimentally confirming both the importance of uncertainty estimation in MBRL and the potential for MBRL to achieve asymptotic performance that is comparable to the best model-free methods at a fraction of the sample complexity.

## 8 Discussion & conclusion

Our experiments suggest several conclusions that are relevant for further investigation in model-based reinforcement learning. First, our results show that model-based reinforcement learning with neural network dynamics models can achieve results that are competitive not only with Bayesian nonparametric models such as GPs, but also on par with model-free algorithms such as PPO and SAC in terms of asymptotic performance, while attaining substantially more efficient convergence. Although the individual components of our model-based reinforcement learning algorithms are not individually new – prior works have suggested both ensembling and outputting Gaussian distribution parameters [Lakshminarayanan et al., 2017], as well as the use of MPC for model-based RL [Nagabandi et al., 2017] – the particular combination of these components into a model-based reinforcement learning algorithm is, to our knowledge, novel, and the results provide a new state-of-the-art for model-based reinforcement learning algorithms based on high-capacity parametric models such as neural networks. The systematic investigation in our experiments was a critical ingredient in determining the precise combination of these components that attains the best performance.

Our results indicate that the gap in asymptotic performance between model-based and model-free reinforcement learning can, at least in part, be bridged by incorporating uncertainty estimation into the model learning process. Our experiments further indicate that both epistemic and aleatoric uncertainty plays a crucial role in this process. Our analysis considers a model-based algorithm based on dynamics estimation and planning. A compelling alternative class of methods uses the model to train a parameterized policy [Ko et al., 2007, Deisenroth et al., 2014, McAllister and Rasmussen, 2017]. While the choice of using the model for planning versus policy learning is largely orthogonal to the other design choices, a promising direction for future work is to investigate how policy learning can be incorporated into our framework to amortize the cost of planning at test-time. Our initial experiments with policy learning did not yield an effective algorithm by directly propagating gradients through our uncertainty-aware models. We believe this may be due to chaotic policy gradients, whose recent analysis [Parmas et al., 2018] could help yield a policy-based PETS in future work. Finally, the observation that model-based RL can match the performance of model-free algorithms suggests that substantial further investigation of such methods is in order, as a potential avenue for effective, sample-efficient, and practical general-purpose reinforcement learning.

## Footnotes

[1]Code available at `https://github.com/kchua/handful-of-trials`

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
