[Supplementary Material · appendix.pdf]

# A    Appendix

## A.1    Well behaved probabilistic networks

An under-appreciated detail of probabilistic networks is how the variance output is implemented with automatic differentiation. Often the real-valued output is treated as a log variance (or similar), and transformed through an exponential function (or similar) to produce a nonnegative-valued output, necessary to be interpreted as a variance. However, whilst this variance output is well behaved at points within the training distribution, its value is undefined outside the trained distribution. In fact, during the training, there is no explicit loss term that regulate the behavior of the variance outside of the training points. Thus, when this model is then evaluated at previously unseen states, as is often the case during the MBRL learning process, the outputted variance can assume any arbitrary value, and in practice we noticed how it occasionally collapse to zero, or explode toward infinity.

This behavior is in contrast with other models, such as GPs, where the variance is more well behaving, being bounded and Lipschitz-smooth. As a remedy, we found that in our model lower bounding and upper bounding the output variance such that they could not be lower or higher than the lowest and highest values in the training data significantly helped. To bound the variance output for a probabilistic network to be between the upper and lower bounds found during training the network on the training data, we used the following code with automatic differentiation:

```
logvar = max_logvar - tf.nn.softplus(max_logvar - logvar)
logvar = min_logvar + tf.nn.softplus(logvar - min_logvar)
var = tf.exp(logvar)
```

with a small regularization penalty on term on `max_logvar` so that it does not grow beyond the training distribution's maximum output variance, and on the negative of `min_logvar` so that it does not drop below the training distribution's minimum output variance.

## A.2    Fitting PE model to toy function

As an initial test, we evaluated all previously described models by fitting to a dataset $\{(x_i, y_i)\}$ of 2000 points from a sine function, where the $x_i$'s are sampled uniformly from $[-2\pi, -\pi] \cup [\pi, 2\pi]$. Before fitting, we introduced heteroscedastic noise by performing the transformation

$$(x, y) \mapsto \left( x, y + \mathcal{N}\left( 0, 0.0225 \left| \sin\left( \frac{3}{2}x + \frac{\pi}{8} \right) \right| \right) \right). \tag{2}$$

The model fit to (2) was shown in Figure 1, but reproduced here for convenience as Figure A.5.

Figure A.5: Our probabilistic ensemble (PE) dynamics model: an ensemble of two bootstraps (for visual clarity, we normally use five bootstraps), each a probabilistic neural network that captures aleatoric uncertainty (in this case: observation noise). Note the bootstraps agree near data, but tend to disagree far from data. Such bootstrap disagreement represents our model's epistemic uncertainty.

## A.3 One-step predictions of learned models

To visualize and verify the accuracy of our PE model, we took all training data from the experiments and visualized the one-step predictions of the model. Since the states are high-dimensional, we resorted to plotting the output dimensions individually, sorting by the ground truth value in each dimension, seen in Figure A.6.

(a) Cartpole dim3 training data aleatoric.

(b) Cartpole dim3 holdout data aleatoric.

(c) Cartpole dim3 training data epistemic.

(d) Cartpole dim3 holdout data epistemic.

Figure A.6: One step predictions of the cartpole angular velocity (velocities are typically harder to predict) after 100 trails of training data. Shown are the prediction indexes, monotonically increase in ground truth output value, with two standard deviations at each output prediction. We see the model is certain (w.r.t. both uncertainty types) where most of the data lies, but less certain in extreme values of data where there are fewer training data.

## A.4 Uncertainty propagation methods

(a) Trajectory sampling (TS1).

(b) Trajectory sampling (TS∞).

(c) Distribution sampling (DS).

(d) Moment matching (MM).

Figure A.7: Different uncertainty propagation methods discussed in Section 5. We show a PE model trained after 100 trials on the cartpole system propagating particles given an action sequence from an intermediate state (pole swinging up) that solves the task.

## A.5 Forward Dynamics Model

Following the suggestion presented in [Deisenroth et al., 2014], instead of learning a forward dynamics in the form $s_{t+1} = f(s_t, a_t)$, we learn a model that predicts the difference to the current state $\Delta s_{t+1} = f(s_t, a_t)$ such that $s_{t+1} = s_t + \Delta s_{t+1}$. Moreover, for states $s_i$ that represent angles, we transform the states fed as inputs to the dynamics model to be $[\sin(s_i), \cos(s_i)]$ to capture the rotational nature of the joint.

## A.6 Experimental setting

For our experiments, we used four continuous-control benchmark tasks simulated via MuJoCo [Todorov et al., 2012] that vary in complexity, dimensionality, and the presence of contact forces (pictured Figure 2). The simplest is the classical cartpole swing-up benchmark ($d_s = 4$, $d_a = 1$). To

evaluate our model with higher dimensional dynamics and frictional contacts, we use a simulated PR2 robot in a reaching and pushing task ($d_s = 14$, $d_a = 7$), as well as the half-cheetah ($d_s = 17$, $d_a = 6$). Each experiment is repeated with different random seeds, and the mean and standard deviation of the cost is reported for each condition. Each neural network dynamics model consist of three fully connected layers, 500 neurons per layer (except 250 for halfcheetah), and swish activation functions [Ramachandran et al., 2017]. The weights of the networks were initially sampled from a truncated Gaussian with variance equal to the reciprocal of the number of fan-in neurons.

## A.7  Additional considerations

**MPC horizon length**: choosing the MPC horizon $T$ is nontrivial: 'too short' and MPC suffer from bias, 'too long' then variance. Probabilistic propagation methods are robust to horizons set 'too long'. This effect is due to particle separation over time (e.g. Figure A.7), which reduces the dependence of actions on *expected*-cost further in time. The action selection procedure then effectively ignores the unpredictable with our method. Deterministic methods have no such mechanism to avoid model bias [Deisenroth et al., 2014], which compounds over longer time horizons, resulting in poor performance if the horizon is set 'too high' as seen in Figure A.8.

(a) Halfcheetah trial 1.

(b) Halfcheetah trial 10.

(c) Halfcheetah trial 40.

(d) Halfcheetah trial 80.

Figure A.8: Effect of MPC horizon on halfcheetah after different amounts of trials. Showing median, and percentile bound 5 and 95, from 5 repeats of experiment.

**MPC action sampling**: We hypothesized the higher the state or action dimensionality, the more important that MPC action selection is guided (opposed to the uniform random shooting method, used by Nagabandi et al. [2017]). Thus we tested cross-entropy method (CEM) and random shooting for various tasks confirming this hypothesis (details Appendix A.8).

**Stochastic systems**: Finally we evaluate how successful probabilistic networks mitigate the detrimental effects of system stochasticity whilst learning to control. We introduced probabilistic networks as a means of capturing aleatoric uncertainty (inherent and persistent system stochasticities). Here we test how well probabilistic networks perform against deterministic networks under stochasticities in the action space. We add Gaussian noise onto the robot's selected action, of standard deviations ranging 0-20% of action ranges permitted by MuJoCo. Figure A.9 shows that probabilistic PE models perform better and more consistently under system noise. Further visualizations are provided in Appendix A.9.

Figure A.9: Modeling aleatoric uncertainty makes MBRL more robust to stochasticity.

**Model accuracy over time**: Figure A.10 shows the evolution of a PE model's accuracy on the halfcheetah as it collects model trails of data (see legend).

(a) Mean squared error.

(b) Negative log likelihood.

Figure A.10: Model accuracy: our PETS dynamics model at trials 10-100 (see legend) make predictions on trajectory seen at each trial (x-axis) and are scored (y-axis) according to mean squared error (left figure) and negative log likelihood (right figure).

## A.8 MPC action selection

We study the impact of the particular choice of action optimization technique. An important criterion when selecting the optimizer is not only the optimality of the selected actions, but also the speed with which the actions can be obtained, which is especially critical for real-world control tasks that must proceed in real time[2]. Simple random search techniques have been proposed in prior work due to their simplicity and ease of parallelism [Nagabandi et al., 2017]. However, uniform random search [Brooks, 1958] suffers in high-dimensional spaces. In addition to random search, we compare to the cross-entropy method (CEM) [Botev et al., 2013], which iteratively samples solutions from a candidate distribution that is adjusted based on the best sampled solutions. To isolate the comparison of optimizers from our dynamics model, we instead use the ground truth dynamics function (the MuJoCo simulator

Figure A.11: Average reward achieved on ground truth dynamics of the half-cheetah (using the MuJoCo simulator itself as ground truth dynamics). The cross entropy method (CEM) optimizer performs significantly better than random shooting sampling. For fair comparison, both use 2500 samples: CEM has five iterations of sampling 500 candidate actions before choosing the elite candidates, whereas random shooting simply sampled 2500 times. Shown is the median performance, with error bars showing the 5 and 95 percentile performance across random seeds.

itself) to evaluate candidate action sequences. The results (Figure A.11) show that using CEM significantly outperforms random search on the half-cheetah task. We use CEM in all of the remaining experiments.

## A.9 Stochastic systems:

In Figure A.12f we compare and contrast the effect stochastic action noise has w.r.t. variable MBRL modeling decisions. Notice methods that PE method that propagate uncertainty are generally required for *consistent* performance.

(a) D-E

(b) P-E

(c) PE-DS

(d) PE-E

(e) PE-TS1

(f) PE-TS∞

Figure A.12: The distribution of cartpole's reward for particular MBRL design decisions in the presence of stochastic system noise (in this case additive noise onto the actions selected by the robot: with standard deviation equal to 10% of each of the action range.)

## A.10 Linear model comparison:

Figure A.13 shows that a linear model is unable to capture the halfcheetah dynamics well enough to control it, and that a nonlinear model is necessary.

Figure A.13: Linear model comparison.

## Footnotes

[2]Such as robotics, where control frequencies below 20Hz are undesirable, meaning that a decision need to be taken in under 50ms.