[Reviews · NeurIPS 2018]

Reviewer 1



This paper describes a model-based reinforcement learning approach which is applied on 4 of the continuous control Mujoco tasks. The approach incorporates uncertainty in the forward dynamics model in two ways: by predicting a Gaussian distribution over future states, rather than a single point, and by training an ensemble of models using different subsets of the agent's experience. As a controller, the authors use the CEM method to generate action sequences, which are then used to generate state trajectories using the stochastic forward dynamics model. Reward sums are computed for each of the action-conditional trajectories, and the action corresponding to the highest predicted reward is executed. This is thus a form of model-predictive control. In their experiments, the authors show that their method is able to match the performance of SOTA model-free approaches using many fewer environment interactions, i.e. with improved sample complexity, for 3 out of 4 tasks. It also performs better than another recent model-based RL approach which uses a deterministic forward dynamics model and a random shooting methods, rather than CEM, for producing candidate actions to be evaluated by the forward model. Quality: The paper performs a thorough experimental evaluation on the 4 tasks and compares against several strong baselines, including model-based and model-free methods. The authors also perform ablation studies which disambiguate the effects of different design decisions, such as probabilistic vs. deterministic outputs of the forward model and single model vs. ensemble of models. These actually show that most of the performance gains come from the use of a probabilistic output, rather than the model ensembling, which somewhat weakens the authors' story in the introduction that modeling both aleatoric and epistemic uncertainty (captured by probabilistic models and ensembling, respectively) is useful, although the gains of using the ensembling are clear for 1 out of 4 tasks. Clarity: The paper is clear and well-written, and results are well-communicated through the figures. Originality: As the authors point out, the components of their method are not novel (probabilistic neural networks, ensembling, using dynamics models for planning), but this particular combination of components applied to these particular tasks is. Significance: It seems to me that the main contribution of this paper is to provide new evidence that with a well-designed experimental setup, model-based RL methods can, indeed, match model-free methods with much better sample complexity. The components of this setup aren't particularly novel, but the authors have done a good job of making their method work empirically which sets a higher standard for future model-based methods. One limitation is that the tasks themselves all have low-dimensional state and actions spaces, which allow the authors to use simple and small fully-connected models together with an explicit paramaterization of the output distribution (as a mean and variance over output states). It is unclear whether the method in its current form would be applicable to more challenging tasks involving higher-dimensional state or action spaces (such as images). Using an explicit paramaterization of a high-dimensional Gaussian is harder to train, and it is not clear if the CEM method they use would scale to high-dimensional action spaces. This may require gradient-based optimization of a paramaterized policy. Overall, I would suggest a weak accept. The contributions of the paper aren't fundamental, but the respectable empirical results help validate the model-based approach to RL, which I believe to be a useful direction overall for adapting RL methods to more practical applications where improved sample complexity is important.

Reviewer 2



The paper is concerned with the topic of increasing data efficiency in reinforcement learning by using an uncertainty aware model. This is a very important topic, in my opinion. The paper presents a method, based on neural network rollouts with stochastic sampling. This method is tested on four different benchmarks and compared to ten alternative approaches. This study is an important contribution, in my opinion. I think the paper would benefit, if it would not claim a too general novelty of the method---there have been very similar methods before, see "Decomposition of Uncertainty in Bayesian Deep Learning for Efficient and Risk-sensitive Learning" Stefan Depeweg et al. arXiv:1710.07283 and "Uncertainty Decomposition in Bayesian Neural Networks with Latent Variables" Stefan Depeweg et al. arXiv:1706.08495 I think the paper should not claim presenting the first neural network based model based RL with uncertainty incorporation, but just report the specific methods and the results, which are a very worthy reference for the ongoing research on the topic of uncertainty aware RL. The paper is well organized and written. These are my concerns: As the paper is on neural network based model based RL, I think in line 22 there should also be mentioned the work on recurrent neural network based RL by Bram Bakker and Anton Schaefer e.g. B. Bakker, The state of mind: Reinforcement learning with recurrent neural networks, Ph.D. thesis, Leiden University, 2004. and Anton Schaefer e.g. A. M. Schaefer, Reinforcement Learning with Recurrent Neural Networks, PhD thesis, University Osnabruck, 2008 In line 27 "MBRL methods ... generally trail behind model-free methods". It should be made clear, that this is a subjective belief, like "our experiments have shown", or "from our experience", "we have observed". Or you should give a reference that supports this rather general claim. Same for the sentence, starting in line 28 "That is, ..." Same for line 37 "they tend to overfit and make poor predictions" and line 38 "For this reason..." Line 80 "In contrast to these prior methods" should be "In contrast to these prior publications" --- the differences are not due to the method, but due to the larger number of benchmarks and methods for comparison in the present paper. In the same sentence "In contrast to these prior methods, our experiments focus on more challenging tasks". It is not obvious, that the benchmarks, used in this paper are "more complex" as the "industrial benchmark" of Depeweg et al. see "Decomposition of Uncertainty in Bayesian Deep Learning for Efficient and Risk-sensitive Learning" Stefan Depeweg et al. arXiv:1710.07283 Please consider, that the separation of aleatoric uncertainty and epistemic uncertainty in Bayesian neural network based model based RL has been analyzed in "Decomposition of Uncertainty in Bayesian Deep Learning for Efficient and Risk-sensitive Learning" Stefan Depeweg et al. arXiv:1710.07283 Post Rebuttal ============ The Author Feedback suggests, that the authors have addressed or will address the concerns which I have expressed w.r.t. too strong claims concerning novelty.

Reviewer 3



The authors investigate the influence of uncertainty prediction in model-based reinforcement learning. They hypothesise that learned transition models should model both aleatoric and epistemic uncertainty to work well in model predictive control (MPC). Probabilistic neural networks capture the former well, but tend to over-fit with few samples and Gaussian processes capture the latter well, but do not scale well. The authors propose to use probabilistic ensemble neural networks and compare their method with a variety of model architectures and propagation methods for MPC. They conclude that both probabilistic networks and ensemble methods significantly improve the performance on MuJoCo tasks. The authors claim that their method reaches the same asymptotic performance as model-free methods with only a fraction of training samples. Including uncertainty estimates into model-based RL is a promising research direction and the authors discuss a variety of potential and existing approaches. The paper is well written and the discussed methods are thoroughly evaluated. The results show significantly improved performance in the harder tasks, like PR2-Reacher and Half-cheetah. As minor criticisms, the authors emphasise the importance of estimating aleatoric and epistemic uncertainty. While the difference is crucial in many ways, the authors never really make use of this difference or discuss how it affects their framework. The authors also claim that their "approach matches the asymptotic performance of model-free algorithms on several challenging benchmark tasks". While this is technically true, the reviewer is missing a discussion about the Half-cheetah task, in which the novel method did not outperform SAC and in which the asymptotic performance of the model-free algorithms differed wildly. Nonetheless, the paper is a very good contribution and should be accepted. The authors should seriously revise the appendix, though. COMMENTS: l.155: the norm in the MSE definition should be squared l.168: the formula refers to the mean, not to the distribution l.185: you state that MPC does not require a planning horizon, but then you introduce on in l.188 and even evaluate the influence in appendix A.7 l.194: you should choose one word in "for in" l.195: "than what" is technically correct, but sounds weird l.288f: it is hard to see a difference between "same bootstrap" and "same bootstrap indexes" l.291: rouge "a" l.308f: you claim that GP-MM outperforms PE-TS1 on cartpole, but in Figure 3, only GP-E outperforms your method. l.327: "aadvanced" Most figures in the appendix are not readable due to tiny fonts It is not obvious how to interpret Figure A.7 A.10 misses any description what the ablation study was about